# The Individual and Combined Entomopathogenic Activity of a *Spodoptera frugiperda* Multiple Nucleopolyhedrovirus and a Type I *Spodoptera frugiperda* Granulovirus on *S. frugiperda* Larvae

**DOI:** 10.3390/v17050674

**Published:** 2025-05-05

**Authors:** Magali Ordóñez-García, Juan Carlos Bustillos-Rodríguez, José de Jesús Ornelas-Paz, Miguel Ángel Salas-Marina, Octavio Jhonathan Cambero-Campos, Carlos Horacio Acosta-Muñiz, David Ignacio Berlanga-Reyes, Claudio Rios-Velasco

**Affiliations:** 1Tecnológico Nacional de México, Campus Cuauhtémoc, Chihuahua, Av. Tecnológico, Cuauhtémoc 31500, Chihuahua, Mexico; mordonez@itcdcuauhtemoc.edu.mx; 2Centro de Investigación en Alimentación y Desarrollo, A.C., Campus Cuauhtémoc, Chihuahua, Av. Río Conchos S/N, Parque Industrial, Cuauhtémoc 31570, Chihuahua, Mexico; jornelas@ciad.mx (J.d.J.O.-P.); cacosta@ciad.mx (C.H.A.-M.); dberlanga@ciad.mx (D.I.B.-R.); 3Unidad Académica Villacorzo, Facultad de Ingeniería, Universidad de Ciencias y Artes de Chiapas, Km 3.0 Carretera Villacorzo-Ejido Monterrey, Villacorzo 30520, Chiapas, Mexico; miguel.salas@unicach.mx; 4Unidad Académica de Agricultura, Universidad Autónoma de Nayarit, Carretera Tepic-Compostela Km 9.0, Xalisco 63155, Nayarit, Mexico; jhony695@uan.edu.mx

**Keywords:** baculoviruses, biological control, fall armyworm, interaction, pest

## Abstract

The bioinsecticidal activity of several doses of a *Spodoptera frugiperda* multiple nucleopolyhedrovirus (SfMNPV-CH-32; LD_10_, LD_50_, and LD_90_) and a Type I *Spodoptera frugiperda* granulovirus (SfGV-CH13; LD_50_ and LD_90_), alone and in co-infection, was evaluated on *S. frugiperda* larvae. In the co-infection assays, one virus was applied at 0 h, and then the second virus was supplied at different times (0, 12, and 24 h) in order to test the effect of the co-infection time on the insecticidal activity of the viruses. The symptoms observed in the co-infected larvae depended on the viral dose supplied at 0 h. The larvae treated with the highest dose (LD_90_) of SfMNPV-CH32 and co-infected with SfGV-CH13 at LD_50_ showed symptoms of nucleopolyhedrovirus infection at 14 days post-infection. The larvae initially infected with the highest dose of SfGV-CH13 (LD_90_) and subsequently co-infected with SfMNPV-CH32 (LD_50_ and LD_10_) showed infection symptoms characteristic of both viruses. The insecticidal activity of SfGV-CH13 and SfMNPV-CH32 alone or in combination depended on the viral doses and the time elapsed between the first and second inoculation. An antagonistic effect was observed for most of the treatments tested. A synergistic effect was observed only in treatment 10, where the larvae were first infected with SfMNPV-CH32 at a high dose (LD_90_) and inoculated 24 h later with SfGV-CH13 (LD_50_).

## 1. Introduction

The fall armyworm (FAW) *Spodoptera frugiperda* (J.E. Smith) (Lepidoptera: Noctuidae) is the main pest of maize (corn; *Zea mays* L.) and other crops in Mexico and other countries, causing crop losses exceeding 30% [1,2,3,4,5]. The control of this insect has been based on chemical insecticides, which cause adverse effects on human health and the environment, with the latter including the elimination of beneficial insects [6]. This has resulted in a search for eco-friendly methods to control the FAW. The use of natural enemies (fungi, bacteria, viruses, nematodes, protozoa, parasitoids, and predator insects) of the FAW as control strategies has advantages over chemical pesticides [7]. The members of the family Baculoviridae are known among entomopathogenic viruses as excellent biocontrol agents of this pest, and have been considered primary components in lepidopteran pest management programs [8]. The *Baculoviridae* family groups four genera, with the *Alphabaculovirus* (nucleopolyhedrovirus; NPV) and *Betabaculovirus* (granulovirus; GV) genera being lepidopteran-specific, and the most widely distributed and studied worldwide [9,10]. The biological activity of NPVs and GVs has widely been documented on many lepidopteran pests [11,12,13,14,15]. NPV isolates are the most used and studied as control agents of the FAW [3,16,17,18,19,20,21]. The widespread use of *S. frugiperda* NPVs is a consequence of their high virulence, infecting several tissues at the same time (e.g., the fat body, tracheal matrix, and epidermis) and causing the death of the insect in a short time (four to six days), even at low doses of occlusion bodies (OBs) [22,23]. The use of Type I *S. frugiperda* GVs as agents for pest control has scarcely been documented [11,24,25]. Type I GVs mainly infect the fat body, causing the slow death of the insect [26]. Studies with other baculoviruses have demonstrated that the co-infection of insects with different types of these viruses can cause effects of potentiation, synergism, and antagonism [27,28,29]. Cuartas-Otálora et al. (2019) [30]. observed that NPV and GV mixtures increased the insecticidal efficacy of one or both. Some types of GVs can easily digest the peritrophic membrane of the insect by the enhancin protein (metalloprotease) present in their OBs, facilitating the access of virions of NPVs and other GVs into the intestinal cells and, consequently, increasing the insecticidal activity of these viruses [31,32]. The current knowledge on the insecticidal activity of NPV and GV co-infections in *S. frugiperda* is scarce and inconsistent [30]. The present study aimed to evaluate the insecticidal activity of a *Spodoptera frugiperda* multiple nucleopolyhedrovirus and a Type I *Spodoptera frugiperda* granulovirus alone and in combination against FAW larvae at different doses and infection times.

## 2. Materials and Methods

### 2.1. Insect Rearing

The FAW larvae were obtained from a laboratory colony which had been reared for 12 months (eight generations) in Cuauhtemoc, Chihuahua, Mexico. The colony was established using larvae collected from a maize plot (latitude 28°12′44″ N, longitude 106°59′45″ W, altitude 2125 m above sea level). The larvae were maintained under controlled conditions (26 ± 2 °C, >70% RH, 12:12 L:D photoperiod) and fed with an artificial diet (Southland Products Inc., Lake Village, AR, USA). The adult moths were fed with a sugar solution and maintained in 15 L cylindrical plastic containers.

### 2.2. Virus Propagation

A *Spodoptera frugiperda* multiple nucleopolyhedrovirus (SfMNPV-CH32) and a Type I *Spodoptera frugiperda* granulovirus (SfGV-CH13) were used in the study. These isolates were described previously [3,24]. The viral isolates were propagated separately in fourth-instar FAW larvae by the droplet feeding method [33]. For the inoculation, 0.5 µL of viral suspension (~10^8^ and ~10^10^ OBs of the NPV and GV, respectively) were supplied to larvae that had been starved for 12 h. The OBs were mixed with Fluorella blue (0.001%, *w*/*v*) and sucrose (10%, *w*/*v*) before use. The infected larvae were individually placed into 29.5 mL plastic cups containing an artificial diet and maintained under the controlled ambient conditions described above. The dead larvae were collected and stored at −20 °C. Then, these larvae were macerated in sterile mortars using sterile distilled water (SDW) containing 1 mg/mL of sodium dodecyl sulfate (SDS). The larval cuticle was removed by filtration through muslin. The filtrates were placed into 10 mL polypropylene tubes and centrifuged (8500× *g*, 4 °C, 10 min). The pellet with the OBs was re-suspended in 10 mL of SDW and stored at −80 °C. Then, the OBs were purified by continuous sucrose gradients (40 and 66%, *w*/*w*) using a gradient former (CBS Scientific, Del Mar, USA, GM 200). Five milliliters of the viral suspension were deposited on the surface of 20 mL of the gradients previously collected into 30 mL polyallomer tubes and ultracentrifuged (40,310× *g*, 4 °C, 1.5 h). The bands containing the OBs were recovered using a Pasteur pipette and placed into 30 mL polypropylene tubes to be washed twice with SDW by ultracentrifugation (40,310× *g*, 4 °C, 40 min). Finally, the pure OBs were re-suspended in 1 mL aliquots of SDW and stored at −80 °C until use.

### 2.3. Bioassays

The insecticidal activity of SfGV-CH13 was evaluated using the median lethal doses of LD_50_ and LD_90_ at 45 days post-infection (dpi), and using the LD_10_, LD_50_, and LD_90_ for SfMNPV-CH32 at 14 dpi. The tested NPV and GV doses (Table 1) were previously determined [3,24].

The co-infection of third-instar (≈7 days after hatching) FAW larvae with the SfGV-CH13 and SfMNPV-CH32 baculoviruses was also evaluated. These bioassays were performed by supplying the lethal doses of both isolates at 0, 12, and 24 h after supplying the first virus (GV or NPV). All bioassays were performed using the droplet feeding method [33]. The FAW larvae used in the bioassays were starved for 12 h, and then supplied with 0.5 μL of the purified viral suspensions, previously counted in triplicate in a Neubauer chamber (Marienfeld, Germany) using a phase-contrast microscope (Carl Zeiss, AxioScope A1; Gottingen, Germany) at 400 and 1000× magnifications for the NPV and GV, respectively [3,24,34]. The OBs were disaggregated by sonication (Branson 1510, CT, USA) for 30 s after being counted, and then mixed with Fluorella blue (0.001%, *w*/*v*) and sucrose (10%, *w*/*v*) before use in the bioassays. The co-infections and single infections contained the same doses of the NPV or GV.

In the case of the co-infection, the inoculation with both isolates (0.5 μL of each viral suspension) at 0 h was performed simultaneously. At 12 or 24 h, the viral suspensions were supplied as described in Table 2. At 12 and 24 h, the larvae infected with the first viral isolate were individually placed into 29.5 mL plastic cups and fed with an artificial diet for 4 h and 12 h, and then starved for 8 h and 12 h, respectively, before being supplied with 0.5 μL of the second viral suspension. At all time points, the larvae were maintained under the controlled ambient conditions described above. Only the larvae consuming the whole inoculum and showing a blue-colored intestinal tract, as confirmed by observation under a stereomicroscope (Carl Zeiss 508, Oberkochen, Germany), were considered in the experiment. Dead larvae were recorded every 8 h until 14 dpi in all treatments. After this day, the larval mortality in the treatments with the GV alone (T1 and T2) and in the mixtures (T7–T20; Table 2) was monitored every 24 h until larval death (~58 d).

A total of 375 larvae were used for monitoring the individual biological activity of the lethal doses indicated above for the NPV and GV isolates, i.e., 75 larvae per treatment (T1–T5; Table 2), and 1125 larvae to monitor the effects of the virus mixtures (T6–T20) in the larvae at different times (0, 12, and 24 h; Table 2). Larvae (75) treated with 0.5 μL of Fluorella blue and sucrose solution were used as an absolute control group (T21; Table 2). The infected larvae were individually placed into 29.5 mL plastic cups, fed with the artificial diet, and maintained under the controlled ambient conditions described above.

### 2.4. Experimental Design and Statistical Analysis

The experiment was conducted under a completely randomized design. In total, 21 treatments were evaluated in triplicate, using 25 larvae per replicate. The obtained data were analyzed by an analysis of variance (ANOVA), and the means were separated by a Tukey’s test (*p* < 0.05). The lethal doses and fiducial limit values were analyzed using log-probit regressions [35]. All data were analyzed using SAS software, version 9.0 [36].

The mortality values obtained 14 days after inoculation were used to determine the possible synergistic or antagonistic effects of SfMNPV-CH32 and SfGVs-CH13 in the mixtures, according to the Koppenhofer and Kaya formula [37].EM = MSfMNPV + MSfGV (1 − MSfMNPV)
where M corresponds to the mortality caused by each virus alone, and EM corresponds to the expected mortality of the combination of both viruses. The resulting values from a chi-square test, χ^2^ = (MC − EM)^2^/EM, estimated using the mortality of the mixture (MC) were compared to the chi-square table value for 1 degree of freedom (χ2 = 3.841) (α = 0.05). If the value exceeded the table value, it was considered as a possible synergistic (positive value) or antagonistic (negative value) effect, depending on the differences in the D = (MC − EM) values.

## 3. Results

### 3.1. The Individual Insecticidal Activity of the Baculovirus

The untreated larvae did not die, and reached the pupal stage 14 days after starting the bioassays. The infection with GVs at LD_50_ and LD_90_ caused a larval mortality of 9.3% and 20% at 14 dpi and 46.7% and 93.2% at 45 dpi, respectively. The larval mortality with SfNPV-CH32 at LD_10_, LD_50_, and LD_90_ was 9.3, 52, and 86.7% at 14 dpi, respectively. The larval mortality percentages observed at the tested doses of both viral isolates were expected, based on the values obtained in the individual bioassays with both viral isolates. The larvae infected with SfGV-CH13 at LD_50_ and LD_90_ survived a long time (28 and 43 dpi, respectively) and showed typical symptoms of GV infection, including the cessation of feeding, slow movement, and a soft and yellowish-white (pale) cuticle (Figure 1a). On the other hand, the larvae infected with the SfNPV-CH32 isolate showed slow movement, vomiting, diarrhea, and lysis of their brown and black cuticle (Figure 1b), and some larvae stopped feeding after ~3 dpi.

### 3.2. The Insecticidal Activity of the Baculovirus Mixtures

The larvae infected with a high dose of NPV (LD_90_) and a medium dose of GV (LD_50_; Table 2) showed typical symptoms of NPV infection at 14 dpi, and after this time, they showed only symptoms of granulovirus infection. On the other hand, the larvae co-infected with the medium dose of the NPV (LD_50_) and the highest dose of the GV (LD_90_) (T11 to T15), as well as the larvae co-infected with the lowest dose of the NPV (LD_10_) and the highest dose of the GV (LD_90_) (T16 to T20), showed symptoms of infection with both viruses at 14 dpi.

The GVs at LD_50_ and LD_90_ are expressed as OBs/larva. These values were obtained from a minimum of six doses (treatments) and estimated at 45 days post-infection. The NPVs at LD_10_, LD_50_, and LD_90_ are expressed as OBs/larva. These values were obtained from a minimum of five doses (treatments) and estimated at 14 days post-infection. The probit regressions were fitted using the SAS program. χ^2^ = goodness of fit test; df = degrees of freedom; and SE = standard error.

The highest larval mortality at 14 dpi was observed in the larvae initially inoculated with the highest dose (LD_90_) of the NPV and, 24 h after, with the lowest dose (LD_50_) of the GV (T10) (Figure 2b). This mortality was similar to that caused by the NPV alone at the highest dose (T5) (Figure 2a) and by the larvae inoculated at the same time with the lowest dose (LD_50_) of the GV and the highest dose (LD_90_) of the NPV (T6).

However, in these three treatments (T5, T6, and T10), the maximum mortality was reached between 8 and 9 dpi (Figure 3a). Retarding for 12 and 24 h, co-infection with the NPV at the highest dose in the larvae previously infected with the lowest dose of the GV (T7 and T8) caused a gradual reduction in the larval mortality, which was of 60 and 54.7% in these treatments, respectively. This trend revealed an apparent effect of antagonism. This effect was observed since 6 dpi, when the larval mortality ranged from 10.7 to 21.8% for T7 and T8. These mortalities were significantly lower than that caused by infection with the NPV alone (82.7%) (T5) (Figure 3a). However, when the highest dose of the NPV was initially (0 h) applied in the larvae and co-infection with the lowest dose of the GV was retarded for 12 and 24 h (T9 and T10), the mortality gradually increased to 69.3 and 93.4%, respectively (Figure 2b).

On the other hand, when the viruses were applied at the same time, but their doses were inverted (T11), i.e., the GV at LD_90_ and the NPV at LD_50_, the mortality was lower than that observed for the co-infection assay involving the GV at LD_50_ and the NPV at LD_90_ (T5), demonstrating the high relevance of the NPV in infection development. All co-infection treatments involving the medium and highest doses (LD_50_ and LD_90_) of the NPV (T6–T15) caused a higher mortality than infection with the GV alone at all tested doses, independently of the infection time with the NPV. Retarding for 12 and 24 h, co-infection with the medium dose of the NPV (LD_50_) in the larvae previously infected with the highest dose of the GV (LD_90_) (T12 and T13) slightly reduced the larval mortality from 46.7 to 37.3 and 32%, as compared to when both viruses were supplied at the same time (T11). However, it was not significant, probably due to the advanced development of the GV infection. The mortality in the treatments involving these concentrations of the viruses (GV at LD_90_ and NPV at LD_50_) was only increased by firstly infecting the larvae with the NPV, and then retarding for 24 h the co-infection with the GV (T15) (Figure 2c).

The lowest mortality values in the co-infection assays were observed with the lowest concentration of the NPV (LD_10_) independently of the supplying time, reaffirming the importance of this virus in infection development (Figure 2d). The larvae co-infection with the highest dose of the GV (LD_90_) and the lowest dose of the NPV (LD_10_) at different inoculation times extended up to ~13 days (55 to 58 dpi) the larval survival time, as compared with the co-infections (~45 dpi) with the medium LD_50_ and highest LD_90_ doses of the NPV and GV. The mortality values were generally similar in the treatments involving this NPV (LD_10_) dose (T16–T20), except when co-infection with the NPV (T18) was retarded for 24 h in the larvae firstly infected with the highest dose of the GV (LD_90_). In this treatment, an increase in the larval mortality was observed, highlighting an increase in mortality of 22.7% compared to the expected value (10%) with the infection with the NPV alone at LD_10_ (Figure 2d and Figure 3c).

## 4. Discussion

No mortality was recorded for the control larvae of *S. frugiperda* (T21), which reached the pupal stage approximately 14 days after inoculation. Similar values were reported by Du Plessis et al. (2020) [38]. On the other hand, the infection symptoms varied with the treatment. The larvae treated only with the GV or NPV developed infection symptoms characteristic of each infecting virus, as reported by others [16,24,25,39]. The larvae treated with the highest dose (LD_90_) of the NPV and co-infected with the GV at LD_50_ (i.e., T5–T10) only showed symptoms of nucleopolyhedrovirus infection at 14 dpi, probably as a consequence of the high dose of the nucleopolyhedrovirus supplied to the larvae. Similarly, Cuartas-Otálora et al. (2019) [30] observed that the co-infection of larvae with high doses of the NPV and low doses of the GV of *S. frugiperda* supplied at the same time predominantly caused symptoms of nucleopolyhedrovirus infection. Similarly, Wennmann et al. (2015) [40] studied the interaction of *Agrotis segetum* nucleopolyhedrovirus B (AgseNPV-B) and an *Agrotis segetum* granulovirus (AgseGV) in neonate common cutworm *Agrotis segetum* larvae. They reported that at low NPV concentrations, the larvae died of granulosis, while at higher NPV concentrations, the larvae mostly died with polyhedra. Nevertheless, the larvae initially infected with the highest dose of the GV (LD_90_) and subsequently co-infected with the NPV (LD_50_ and LD_10_) at different times showed symptoms of infection characteristic of both viruses. In our study, the inoculation order defined the success of each viral isolate, as reported by others [41].

Different interaction effects were observed in the experiment, depending on the dose and inoculation time of each virus. The expected mortality values in most combinations suggested an antagonistic effect, except in treatment 10, where a synergistic effect was evident (Table 2). This effect of antagonism was observed in T7, T8, T12, and T13, causing a decrease in the larval mortality, probably because in these treatments, the replication of the GV started at 12 and 24 h compared to that of the NPV, causing a competition among the viruses for the same replication site [41].

This antagonistic effect has been scarcely documented in studies on the interactions of baculoviruses [29,42]. This effect has been widely documented in interactions between baculoviruses and parasitoid insects, finding that both compete for host resources and especially for replication sites, with the first of them to infect or parasitize the host causing the predominant effect [41]. Hackett et al. (2000) [29] documented an antagonistic impact on *H. zea* larvae co-infected with a *Helicoverpa armigera* granulovirus (HaGV-9.6 × 10^6^ OBs per cup) and a *Helicoverpa zea* nucleopolyhedrovirus (HzSNPV-10^3^ OBs per cup). They attributed the antagonism to a competition between both viruses for the resources of the host insect and the HaGV protein inhibiting the HzSNPV replication. They observed that the antagonistic effect was still present even when the HaGV was supplied 36 h after the HzSNPV, a time at which the nucleopolyhedrovirus should fully be established in the host and penetrate the peritrophic membrane. In our study, the antagonism effect was not observed when the GV was supplied 24 h after the delivery of the NPV (i.e., T10), probably due to the advanced replication level of the NPV, since nucleopolyhedroviruses replicate their DNA between 6 hpi and 18 dpi [42].

Several studies using two different baculoviruses have demonstrated increased viral mortality or synergistic effects on the target larvae compared to single infections [43]. A synergistic effect was observed in treatment 10, where the larvae were first supplied with SfMNPV-CH32 (LD_50_), and 24 h later, they were co-infected with SfGV-CH13 (LD_90_). An increase in the insecticidal activity of the NPV was observed in these treatments due to the co-infection. The synergism could be given due to the advantage of the NPV in the infection time, although the GV probably favored the NPV infection, as some granuloviruses can rapidly degrade the peritrophic membrane. Although in this study, the synergistic effect was observed when the viruses were placed 24 h apart, Barrera et al. (2021) [44] found a synergistic effect when both viruses (*Spodoptera ornithogalli* nucleopolyhedrovirus (SporNPV) and granulovirus (SporGV)) were co-inoculated into *S. ornithogalli* and *S. frugiperda* larvae [44]. Guo et al. (2007) [45] observed increases in the insecticidal capacity (virulence) of a *Spodoptera litura* nucleopolyhedrovirus (SlNPV) on *S. litura* larvae when the larvae were co-infected with a granulovirus of *Xestia c-nigrum* (XcGV), noting the rupture of the peritrophic matrix in the larvae treated with the XcGV alone or with both viruses in co-infection. Cuartas-Otálora et al. [30] also documented an enhancement in the insecticidal activity (~27% larval mortality) after the co-infection of second-instar FAW larvae with the isolate SfCOL of a *S. frugiperda* multiple nucleopolyhedrovirus (SpfrMNPV) and low doses of the isolate VG008 of a *S. frugiperda* granulovirus (SpfrGV). They attributed this enhancement to proteins from the OBs of *S. frugiperda* granulovirus VG008, which caused holes in the peritrophic membrane, facilitating the access of NPV virions. On the other hand, an increase in the larval survival time was observed in some treatments (T17–T20) where the larvae were supplied either 12 or 24 h later with the lowest dose of the NPV (LD_10_) or 12 or 24 h later with the highest dose of the GV (LD_90_). These viral proportions might favor a higher number of OBs, since the larvae survived for more extended periods and probably reached a larger size, as reported previously [46]. Similar results were reported by Hackett et al. (2000) [29], who found that the survival time of *H. zea* larvae increased as the dose of the GV was increased. The increase in larval survival time could be associated with a sublethal effect of the Type I granulovirus, causing changes in the physiological processes related to developmental stages, development times, and larval death. However, the mechanisms driving the effects of these sublethal doses are poorly understood, and in most cases, the effects have been related to the hormonal and enzymatic changes associated with viral infections [47].

## 5. Conclusions

The results demonstrated that the inoculation time and dose influenced the insecticidal activity of SfMNPV. An antagonistic effect was observed in most treatments. The greatest antagonistic effect occurred when caused by a retardation of 12–24 h in the supply of higher doses (LD_50_ and LD_90_) of the SfMNPV-CH32 isolate to larvae previously infected with high doses (LD_50_ and LD_90_) of SfGV-CH13. A synergistic effect was caused by a high dose (LD_50_) of SfGV-CH13 supplied at 24 h to larvae previously treated with a high dose (LD_90_) of SfMNPV. High doses (LD_90_) of the GV in combination with low doses (LD_10_) of the NPV increased by ~3–4 times the larval development time.

## Figures and Tables

**Figure 1 viruses-17-00674-f001:**
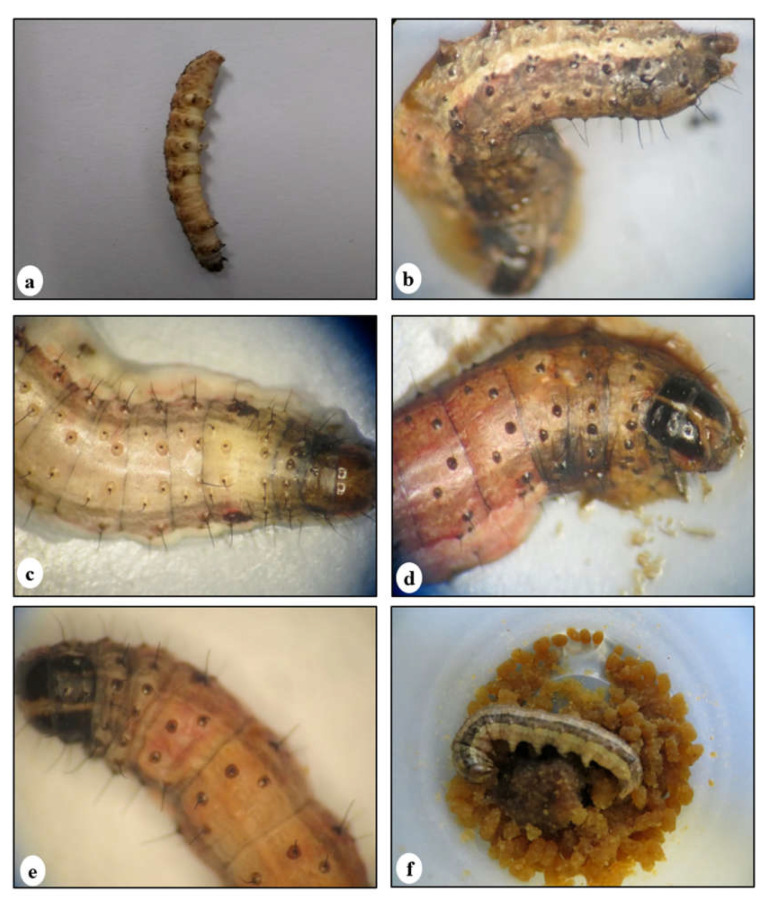
The appearances of FAW larvae infected with the tested isolates alone and in mixtures. (**a**) A larva infected with the SfGV-CH13 isolate alone at a dose of 4.3 × 10^5^ OBs/larva at 11 days post-infection (dpi) without a change in coloration or a rupture of the integument; (**b**) A larva infected with the SfNPV-CH32 isolate alone at a dose of 6.6 × 10^3^ OBs/larva at 6 dpi; the dark color of the fat body indicates the rapid development of NPV infection and the liquefaction of the cadaver; (**c**) A larva infected with the GV and NPV at the same time at doses of LD_50_:LD_90_ at 6 dpi; (**d**) A larva infected with the GV and NPV at the same time at doses of LD_90_:LD_50_ at 6 dpi; the different light color of the fat body in (**c**,**d**) indicates mainly the symptoms of NPV infection, with the liquefaction of the cadaver. (**e**) Larvae infected with the GV and NPV at the same time at doses of LD_90_:LD_10_ at 6 dpi, the light color of the fat body indicates a change in coloration, but no rupture of the integument. (**f**) An un-infected fifth-instar larva (control).

**Figure 2 viruses-17-00674-f002:**
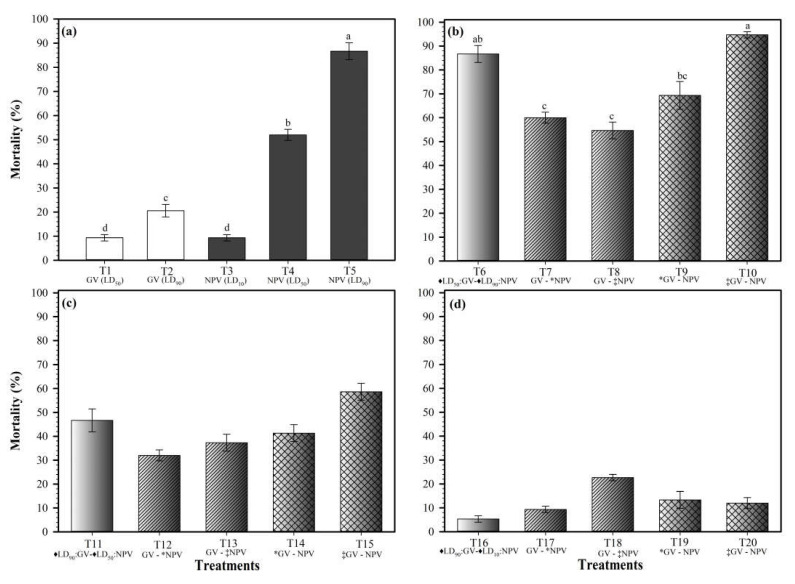
The larval mortality caused by the isolates of SfGV-CH13 (GV) and SfNPV-CH32 (NPV) at different concentrations (LD_10,_ LD_50_, and LD_90_), alone (**a**) and co-infecting (**b**–**d**) the larvae after 0, 12, and 24 h of delivery of the first virus. The data were taken at 14 days post-infection. ^♦^, the viruses were delivered at the same time (0 h), *, the second virus was delivered 12 h after infection with the first isolate. ^‡^, the second virus was delivered 24 h after infection with the first isolate. Equal literals on the standard error bars indicate no statistical differences between the isolates at the same dose, according to a Tukey’s test (*p* < 0.05).

**Figure 3 viruses-17-00674-f003:**
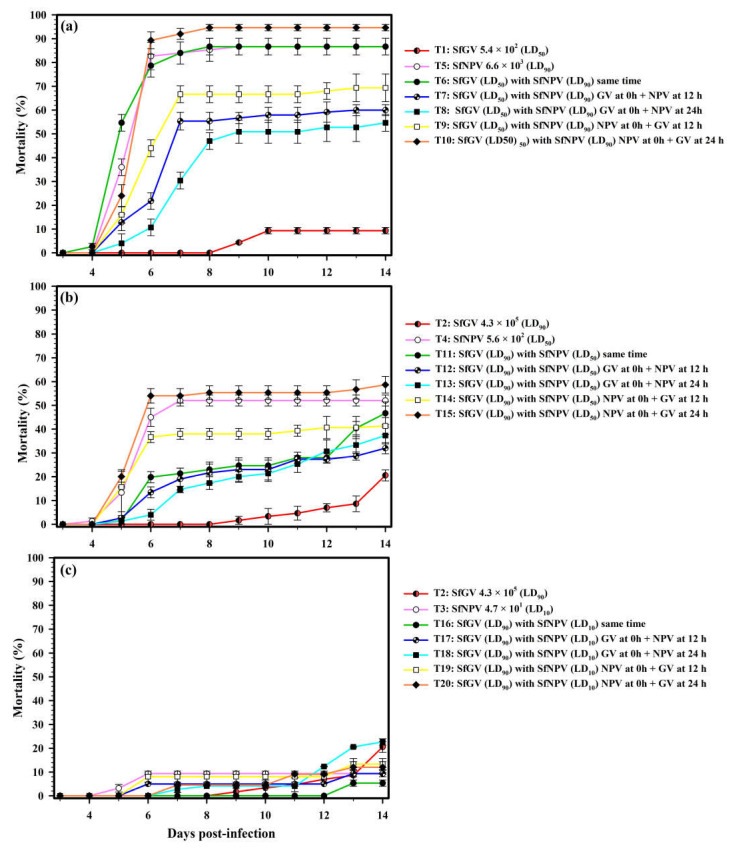
Larval mortality caused by isolates of SfGV-CH13 and SfNPV-CH32 alone and in co-infection from 3 to 14 days post-infection. The right side of the figure shows the treatments evaluated and compared in each graph (**a**–**c**).

**Table 1 viruses-17-00674-t001:** Lethal doses of tested granulovirus and nucleopolyhedrovirus isolates against third-instar *S. frugiperda* larvae.

Isolate	Lethal Doses	Doses (OBs/Larva)	Fiducial Limits (95%)		χ^2^	df	Slope ± (SE)	Intercept ± (SE)
			Lower	Upper				
SfGV-CH13	LD_50_	5.4 × 10^2^	3.1 × 10^2^	9.5 × 10^2^	4.5	4	0.4 ± 0.04	−1.2 ± 0.13
	LD_90_	4.3 × 10^5^	1.3 × 10^5^	2.3 × 10^6^	4.5	4	0.4 ± 0.04	−1.2 ± 0.13
SfNPV-CH32	LD_10_	4.7 × 10^1^	2.6 × 10^1^	6.7 × 10^1^	4.9	3	1.2 ± 0.08	−3.3 ± 0.23
	LD_50_	5.6 × 10^2^	4.3 × 10^2^	7.4 × 10^2^	4.9	3	1.2 ± 0.08	−3.3 ± 0.23
	LD_90_	6.6 × 10^3^	4.4 × 10^3^	1.1 × 10^4^	4.9	3	1.2 ± 0.08	−3.3 ± 0.23

**Table 2 viruses-17-00674-t002:** Treatments of third-instar *S. frugiperda* larvae with SfGV-CH13 and SfNPV-CH32 isolates at 14 days post-infection.

	Dose OBs/Larva		Inoculation of Viral Isolates	Interaction Type
Treatment	SfGV-CH13	SfNPV-CH32		
T1	5.4 × 10^2^ (LD_50_)	--	Alone	--
T2	4.3 × 10^5^ (LD_90_)	--	Alone	--
T3	--	4.7 × 10^1^ (LD_10_)	Alone	--
T4	--	5.6 × 10^2^ (LD_50_)	Alone	--
T5	--	6.6 × 10^3^ (LD_90_)	Alone	--
T6	**^♦^** LD_50_	**^♦^** LD_90_	Same time	Antagonism
T7	LD_50_	* LD_90_	GV at 0 h + NPV at 12 h	Antagonism
T8	LD_50_	^‡^ LD_90_	GV at 0 h + NPV at 24 h	Antagonism
T9	* LD_50_	LD_90_	NPV at 0 h + GV at 12 h	Antagonism
T10	^‡^ LD_50_	LD_90_	NPV at 0 h + GV at 24 h	Synergism
T11	**^♦^** LD_90_	**^♦^** LD_50_	Same time	Antagonism
T12	LD_90_	* LD_50_	GV at 0 h + NPV at 12 h	Antagonism
T13	LD_90_	^‡^ LD_50_	GV at 0 h + NPV at 24 h	Antagonism
T14	* LD_90_	LD_50_	NPV at 0 h + GV at 12 h	Antagonism
T15	^‡^ LD_90_	LD_50_	NPV at 0 h + GV at 24 h	Antagonism
T16	**^♦^** LD_90_	**^♦^** LD_10_	Same time	Antagonism
T17	LD_90_	***** LD_10_	GV at 0 h + NPV at 12 h	Antagonism
T18	LD_90_	**^‡^** LD_10_	GV at 0 h + NPV at 24 h	Antagonism
T19	***** LD_90_	LD_10_	NPV at 0 h + GV at 12 h	Antagonism
T20	**^‡^** LD_90_	LD_10_	NPV at 0 h + GV at 24 h	Antagonism
T21	Un-infected larvae		Control	

**^♦^**, supplied at the same time. *, supplied 12 h after the infection with the first isolate. ^‡^, supplied 24 h after the infection with the first isolate. -- There was no interaction because only one isolate was evaluated.

## Data Availability

The raw data supporting this article will be made available by the authors on request.

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
