# Peer review of "The Individual and Combined Entomopathogenic Activity of a Spodoptera frugiperda Multiple Nucleopolyhedrovirus and a Type I Spodoptera frugiperda Granulovirus on S. frugiperda Larvae"

_viruses, 2025, doi:10.3390/v17050674_

Round 1

Reviewer 1 Report

Comments and Suggestions for Authors

My comment are included in the attached PDF file.

Reviewer 2 Report

Comments and Suggestions for Authors

In this study, the authors characterized the interaction between an alphabaculovirus (SfMNPV) and a betabaculovirus (SfGV) in co-infected larvae of Spodoptera frugiperda, the fall armyworm.  The authors evaluated mortalities in larvae co-infected with different doses of each virus (LD10, LD50, LD90) and also compared mortalities in larvae that were infected simultaneously with mortalities in larvae with staggered infections (e.g. infection with one virus, then infection with a second virus 12 or 24 hr later). 

The authors characterized the interactions between the two viruses in different infection regimes as synergistic or antagonistic, and also had a third interaction category which they referred to as “neutralism” by which they probably meant an additive interaction.  However, their designation of the type of interaction involved appeared to be based solely on comparing the relative mortalities in the different treatments.  This is not sufficient.  There is a statistical analysis used for determining if an interaction is synergistic, antagonistic, or additive, which is described in Koppenhofer & Kaya, Biological Control 8, 131–137 (1997), with an application to baculoviruses described in reference [44] of the authors’ manuscript.

Furthermore, the authors used mortality recorded at 14 days p. i. to assess interactions.  Since the analysis for identifying interactions requires both expected and observed mortalities of the different viruses, the authors should use the mortalities recorded for the entire duration of the bioassays, e.g. after all larvae have either pupated or died, for their analysis.  Lines 122-124 indicate that they have mortality data for the entire duration of the bioassays, so this should not be a problem.

Other comments:

  • Lines 42, 44-45: The authors should consult Archives of Virology (2022) 167:1231–1234, https://doi.org/10.1007/s00705-021-05323-4 “Differentiating between viruses and virus species by writing their names correctly” (written and published by the executive committee of the ICTV) for guidance on how to properly write the names of virus and virus taxa.
  • Lines 102-103: The authors should include the age of the larvae used for bioassays in terms of days post-hatch, as larvae are likely to exist as any given instar >1 day.
  • Lines 115-117: (a) Did the larvae that were infected with a second virus 24 hr after the first undergo a molt to 4th instar prior to, or soon after, the second infection? This may impact the observed mortality. (b) Please clarify/confirm that the 0.5 microliter volumes used for co-infections contained the same dose of NPV or GV as the 1.0 microliter volumes used for single infections with either NPV or GV – for example, 0.5 and 1.0 microliter droplets of SfMNPV both contained an LD10, LD50, or LD90 dose of SfMNPV OBs.
  • Lines 145-150: While these details are useful, it seems like the primary feature distinguishing an NPV infection from a GV infection would be death by 14 days p. i. with cuticular rupture vs. death occurring long after the pupation of uninfected controls.
  • Table 1: The Chi-square, slopes and intercepts for the GV dose-response data have the same values as those of the NPV dose-response data. Surely this is an error.
Comments on the Quality of English Language

The quality of English needs some improvement.

Round 2

Reviewer 1 Report

Comments and Suggestions for Authors

They have improved the English where I commented. I hope they did the same throughout the manuscript. However, the short time (3-days that includes a weekend) to return a critique of the entire manuscript does not allow sufficient time to produce a constructively critical analysis and review. 

Reviewer 2 Report

Comments and Suggestions for Authors

Two comments on this revision:

1) When writing a virus name, no part of a virus name is italicized.  Hence, in the names of the viruses "Spodoptera frugiperda granulovirus" and "Spodoptera frugiperida multiple nucleopolyhedrovirus", the "Spodoptera frugiperda" part is not italicized, even though it would normally be italicized when referring to the lepidopteran species.

2) The explanation for not conducting a chi-square test is not clear.  "MC" in the formula "χ2 = (MC-EM)2/EM" refers to observed mortality of the virus mixtures.  It is not necessary to know the lethal median dose of the virus mixtures, only the observed mortalities of the mixtures and the expected mortalities of the mixtures calculated from the observed mortalities of the individual viruses at the various doses they used in the mixtures.  The chi-square statistic for df=1 and a significance level α=0.05 is 3.841; a calculated chi-square higher than 3.841 provides the necessary statistical support for their claims of synergism and antagonism.

Comments on the Quality of English Language

I presume MDPI will do English language copy-editing on the manuscript.
